

# Unconscious information processing of table tennis athletes in a masked priming paradigm: an event-related potentials (ERP) study

Fanying Meng[1], Lijiao Chen[2], Chun Xie[3], Jiadong Zheng[1], Ning Chen[1], Fanghui Qiu[4] and Jiaxian Geng[1]

[1] Institute of Physical Education, Huzhou University, Huzhou, China
[2] Huzhou Sports School, Unaffiliated, Huzhou, China
[3] Department of Physical Education, Shanghai Jiao Tong University, Shanghai, China
[4] Department of Physical Education, Qingdao University, Qingdao, China

Corresponding author
Jiaxian Geng, 02626@zjhu.edu.cn

## ABSTRACT

**Background:** Unconscious information processing is enhanced among athletes for sports-specific contexts. Whether this enhancement is transferable to general contexts is unknown. This study explored unconscious information processing and brain activity in highly trained table tennis athletes and non-athletes in general contexts.

**Methods:** Twenty table tennis athletes (six females, mean age = 20.38 ± 1.28, mean ± standard error) and 21 aged-matched college students (eight females, mean age = 19.81 ± 1.29) were recruited for this study. Each participant first performed a masked priming task. In this task, a prime stimulus (arrows pointing left or right) was presented, followed by a visual mask (arrows pointing in both directions) and then a target stimulus, the target stimulus consisted of arrows pointing in the same direction as the prime for congruent stimuli or in the opposite direction for incongruent trials, while the P3 component of the event-related potential was simultaneously recorded in the brain. As a control, participants then performed a prime identification task (the subjective threshold test and the objective threshold test) to determine whether they could consciously detect the priming arrows. Reaction times, error rates, P3 latency and P3 peak amplitude were analyzed to examine the unconscious information processing of table tennis athletes in general contexts.

**Results:** Participants responded with the direction of the target arrow and were not consciously aware of the priming stimulus. Athletes responded faster in comparison of non-athletes. Athletes and non-athletes responded faster and committed fewer errors in incongruent *vs.* congruent conditions. In addition, the years of table tennis training were negatively correlated with the magnitude of negative compatibility effect. Both groups displayed longer P3 latencies, a measure of inhibitory control, in the incongruent *vs.* congruent trials. However, athletes displayed higher P3 peak amplitudes, reflecting larger attention resource input, and longer P3 latencies than non-athletes in central brain sites.

**Conclusion:** Unconscious information processing among table tennis athletes is not prominent in general contexts, but may be limited to the sports-specific context or more complex cognitive tasks.

## INTRODUCTION

In modern competitive sports, the speeds at which athletes and objects move exceed the human perception threshold. Accumulating studies have shown that unconscious information processing is thus a necessary skill of athletes and has become a key factor in determining an athlete's performance in competitions (*Jiang, Xie & Li, 2021*; *Mao et al., 2022*; *Meng et al., 2019a*). Athletes from open-skill sports are able to respond successfully to the actions of their teammates or opponents during training or competition, despite their inability to recognize any particular trigger stimuli, such as extremely fast serves or rapid action sequences (*Kibele, 2006*). Table tennis is a sport characterized by rapid tempo, imposes extraordinary demands on the perceptual and cognitive faculties of its athletes. Table tennis athletes are required to swiftly assimilate visual data, anticipate the trajectory and spin of the ball, and carry out exacting motor actions within mere milliseconds (*Chen et al., 2022*; *Yao et al., 2020*). These demands are thought to promote specialized neural adaptations, such as enhanced visuomotor coordination (*e.g.*, improved visual-motor integration) and refined unconscious information processing (*e.g.*, more efficient unconscious inhibitory control), which may have far-reaching implications beyond the realm of athletics (*Meng et al., 2019b*; *Yao et al., 2024*). Previous research has demonstrated that table tennis athletes exhibited advanced unconscious information processing within the sport-specific stimulation scenario (*Meng et al., 2019a*; *Meng, Geng & Li, 2022*). However, it remains unclear whether such superior processing capabilities extend to general stimulation scenario. Therefore, the aim of this study is to investigate the influence of motor experience on unconscious information processing within general stimulation scenarios.

The masked priming paradigm has been widely adopted to investigate the neural mechanisms underlying individuals' response to priming stimuli in an unconscious state (*Rohr & Wentura, 2021*; *Shi et al., 2022*; *Van den Bussche, Van den Noortgate & Reynvoet, 2009*). The unconscious state is typically created by shortening the time of the prime presentation and inserting a visual masked stimulus before the target stimulus (*Cayado, Wray & Stockall, 2023*; *Huang et al., 2024*; *Ortells et al., 2016*). The negative compatibility effect (NCE) constitutes a pivotal phenomenon within the masked priming paradigm, where participants respond faster and commit fewer errors when the prime and target belong to different responses compared with when they belong to the same responses (*Eimer & Schlaghecken, 1998*; *Panis & Schmidt, 2016*; *Wang & Zhang, 2014*). Investigating the NCE enables a profound understanding of the mechanisms of unconscious information processing.

In a masked priming paradigm conducted with experts in specialized contexts (*e.g.*, experienced typists, expert chess players), these individuals not only demonstrated faster reaction times but also exhibited enhanced unconscious information processing compared to the control group (*Heinemann et al., 2010*; *Kiesel et al., 2009*). This finding indicates that

experience or practice may serve as a necessary prerequisite for unconscious information processing. Such unconscious processing plays a particularly critical role in domains that demand swift responses and high adaptability, as exemplified by open motor skills. Open motor skills are characterized by fast offensive and defensive transitions and complex and varied techniques and tactics. Previous studies have found that the decisions that most experienced athletes make in sports scenarios under time pressure are unconscious and automatic (*Gilovich, Griffin & Kahneman, 2002*; *Raab, 2003*; *Wang, 2004*; *Yang & Wang, 2019*). Utilizing a masked priming paradigm, recent studies demonstrated a significant unconscious priming effect—termed the positive compatibility effect (PCE)—among table tennis and taekwondo athletes. This effect manifested as reduced response times and error rates in the congruent condition compared to the incongruent condition, a phenomenon that was not observed in non-athletes (*Güldenpenning et al., 2015*; *Meng et al., 2019a*), these findings indicate that athletes can process motor information quickly and efficiently, even under unconscious condition. *Meng, Geng & Li (2022)* investigated brain activity characteristics in table tennis athletes during unconscious information processing through a masked priming paradigm. Their results suggest that table tennis athletes exhibit faster processing speeds and more effective unconscious information processing compared to non-athletes, likely attributable to the integrated regulation of the ventral and dorsal pathways. Athletes exhibit enhanced motor skills and executive control after extensive specialized training, but whether such training-induced adaptations generalize to broader contexts remains unclear (*Chang et al., 2017*). While prior studies have confirmed that specialized training improves unconscious information processing in sports-specific tasks, its generalization to non-sport domains is yet to be determined. To address this gap, the present study focuses on unconscious information processing in general contexts among table tennis athletes.

Event-related potentials (ERP) have been widely used to investigate the mechanisms underlying cognitive processes due to its high temporal resolution and precise synchronization with neural activity. In the masked priming task, the negative compatibility effect reflects inhibition processes that occur when the target response is congruent with the prime response (*Wang, Jiao & Zhang, 2016*). The P3 component of the ERP, typically recorded in the central-parietal region of the brain, is related to inhibitory control and attentional processes (*Ziri et al., 2024*). Specifically, P3 latency indicates the completion of inhibitory control, whereas P3 amplitude reflects both inhibitory control and the allocation of attentional resources, with higher amplitudes signifying greater attentional resource investment (*Jin, Li & Tao, 2015*; *You et al., 2018*). Analyses of P3 latency and amplitude reveal distinct neural signatures of unconscious information processing, contributing to a deeper understanding of how unconscious information is regulated through inhibitory control and how attentional resources are allocated in table tennis athletes.

To summarize, the current study aimed to explore the characteristics of unconscious information processing and associated brain activity among table tennis athletes in the general contexts, employing a masked priming paradigm combined with ERP measurements. Based on previous studies indicating that specialized training enhances

unconscious information processing (*Güldenpenning et al., 2015*; *Meng et al., 2019a*), we hypothesized that both table tennis athletes and non-athletes would exhibit negative compatibility effects, with table tennis athletes demonstrating better performance compared to non-athletes.

## MATERIALS AND METHODS

### Participants

In alignment with the findings of *Meng et al. (2019a)* regarding the interaction between expertise and response congruency, we calculated the sample size using G*Power 3.1 ($\alpha = 0.05$, power = 0.80, effect size = 0.19). The results showed that at least 40 participants were required.

All participants were college students recruited through posters displayed on campus and were allocated to an athlete group or a non-athlete group according to whether they had professional training in table tennis. The athlete group was composed of 20 table tennis athletes (six females, mean age = 20.38 ± 1.28, mean±standard error) who reached the national second-level athlete standard or above, as certified by the Chinese Table Tennis Association. All of them had at least 7 years of professional table tennis training and maintained more than 7 h table tennis training per week. The non-athlete group was composed of college students (eight females, mean age = 19.81 ± 1.29) who had no table tennis or other sports training experience (Table 1). Participants with equipment malfunction or those who withdrew from the study prematurely were excluded. There was no significant difference in sex, age, body mass index, educational level and other basic information between the two groups except for the years of professional training in table tennis. The inclusion criteria for participants were as follows: all individuals were right-handed, had normal or corrected-to-normal visual acuity, and were in good general health with no history of psychiatric disorders, neurological conditions, or substance abuse. All participants provided written informed consent and received a small financial compensation for their participation. This study protocol received approval from the Ethics Committee of Huzhou University (No. 20200528-SGY13).

### Stimuli and equipment

Double arrows (or two chevrons) both pointing to the left or to the right were selected as primes and targets (Fig. 1). Double arrows pointing left and right were superimposed on each other to form a mask (*Eimer & Schlaghecken, 1998*). All stimuli were presented on a gray background with a size of 2.5 × 5.84 cm and subtended a visual angle of 2.39° horizontally and 5.58° vertically from a viewing distance of 60 cm.

A Dell computer with a 16-inch display (frequency 60 Hz, resolution 1,366 × 768) was used for stimulus presentation. The E-prime 2.0 software package (Psychology Software Tools, Pittsburgh, PA, USA) was used for stimulus presentation and response recording.

### Study design

This study employed a 2 (group: athletes *vs*. non-athletes) × 2 (condition: congruent *vs*. incongruent) mixed factorial design, with group as the between-subjects factor and
**Table 1  Participant demographic characteristics (mean ± standard error).**

| Group | No. | Sex (No.) male/female | Age (years) | Years of training (years) | Body mass index (kg/m$^2$) |
|---|---|---|---|---|---|
| Athletes | 20 | 14/6 | 20.38 ± 1.28 | 7.48 ± 1.34 | 21.77 ± 2.98 |
| Non-athletes | 21 | 13/8 | 19.81 ± 1.29 | 0 | 22.15 ± 3.70 |

condition as the within-subjects factor. The dependent measures comprised both behavioral performance (reaction time and error rate) and electrophysiological indices (P3 peak amplitude and latency).

## Task and procedure

Participants sat comfortably 60 cm from a computer screen in an acoustically shielded room. Participants were asked to fixate on the center of the screen throughout the experiment and to respond to the direction of the target double arrow as quickly and accurately as possible. When the double arrows pointed left, the participants used their left index finger to press the "f" key on a computer keyboard; when the double arrows pointed right, the participants used their right index finger to press the "j" key.

A schematic of the task procedure is given in Fig. 1. First, a central fixation cross "+" was displayed for 750 ms. A blank screen was then displayed for 200 ms. The prime stimulus was next displayed for 16 ms, followed by a mask stimulus for 33 ms. The target stimulus was then displayed for 100 ms, followed by another blank screen displayed for 1,000 ms. Participants were instructed to react to the target within 1,100 ms. The intertrial interval varied randomly from 1,000 to 1,500 ms. During this interval, the screen remained blank until the appearance of the fixation cross, which marked the onset of the next trial.

All trails were divided into two conditions (congruent and incongruent) based on the relationship between the prime and target stimuli. Half the trials were congruent (both the prime stimulus and the target stimulus pointed left or pointed right) and the other half were incongruent (prime stimulus pointed left and target stimulus pointed right or vice versa).

Before the formal experiment, participants were required to complete a practice session to familiarize themselves with the experimental task. The practice block contained four possible combinations of the prime and target stimuli for a total of 24 trials. After the practice, participants performed the formal experiment, which consisted of three blocks of 168 trials.

After the formal experiment, participants were informed of the presence of the prime stimulus and then completed the prime identification control task to assess whether the prime stimuli were consciously perceived or not. The prime identification control task consisted of two parts: the subjective threshold test and the objective threshold test. The subjective threshold test was performed first. It was conducted as an interview in which participants were asked questions about the prime stimulus, such as, "Did you see a shape that briefly flashed before the target stimulus?" or "What do you think the shape that you saw was?" The subjective threshold test followed. The number of trials, the procedure, and the response rules for the subjective threshold test were identical to the formal experiment.

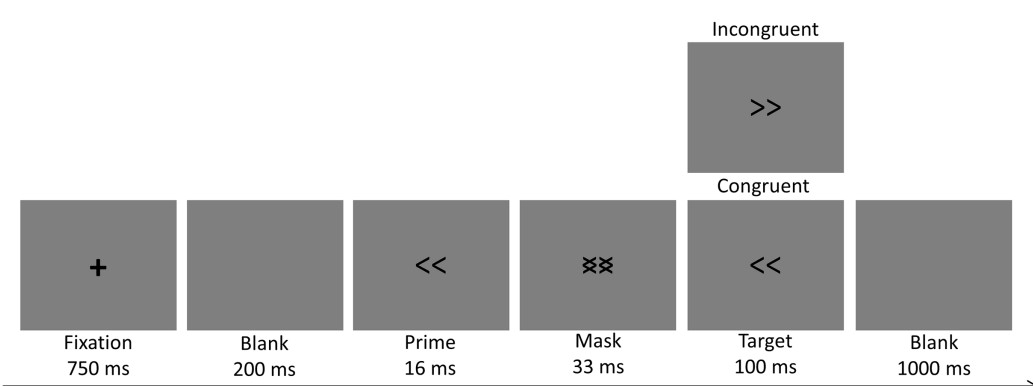

**Figure 1 Sequence of events in each trial of the masked priming task.**

Participants were asked to judge the direction that the arrows of the prime stimulus were pointing while ignoring the target stimulus. The prime identification control task focused on accuracy rather than speed; thus, the response times to the prime stimuli were not recorded.

## Electroencephalography recording and analysis

Brain electrical activity was continuously recorded using 64 Ag/AgCl electrodes arranged according to the International 10–20 system (antiCAP, Brain Products GmbH, Munich, Germany), connected to an ActiCHamp amplifier (Brain Products GmbH, Munich, Germany). The electroencephalography (EEG) data were sampled at 500 Hz and filtered online with a bandpass filter of 0.01–100 Hz. The FCz site served as the reference electrode, and the AFz site served as the ground electrode. The horizontal electrooculography (HEOG) was recorded from an electrode placed 1 cm lateral to the right eye, and the vertical electrooculography (VEOG) was recorded from an electrode placed 1 cm below the left eye. The electrode impedance was kept less than 5 kΩ during the entire acquisition.

## Data analysis

The signal detection measure $d'$ was used to assess the prime visibility (*Wickens, 2001*). A correct response to the target in the congruent trials was considered a hit, and an incorrect response to the target in the incongruent trials was considered a false alarm (*Geng et al., 2020*; *Güldenpenning et al., 2011*). The hit and false-alarm rates of each participant were calculated. We calculated $d'$ as z (hit rates) − z (false-alarm rates), where z represents standard score. One-sample t-tests were performed to determine whether the identification performances of the two groups were at the level of chance, that is, at 50% (which was indicated by $d' = 0$). In addition, independent t-tests were performed to compare the identification rates between athletes and non-athletes.

In order to limit the influence of incorrect and extreme values, incorrect and missed trials (3.75%) and response times (RTs) greater and less than two standard deviations (15.91%) were excluded. The mean RTs for correct responses and the mean error rates (ERs) of each participant and each experimental condition were calculated. The Inverse

Efficiency Score (IES), calculated as reaction time divided by accuracy, served as a composite measure of performance, balancing speed and accuracy. To analyze the behavioral data, two-way repeated-measures analyses of variance (ANOVAs) were used to examine the RTs and ERs, with groups (athlete *vs.* non-athlete) as the between-subjects factor and response congruency (congruent *vs.* incongruent) as the within-subjects factor.

The magnitude of NCE was quantified as reaction time in congruent trials minus reaction time in incongruent trials ($RT_{congruent} - RT_{incongruent}$). The association between the years of training and the magnitude of NCE among table tennis athletes was examined through Pearson's correlation analysis.

EEG data were preprocessed offline with Brain Vision Analyzer software (version 2.0, Brain Products, Gilching, Germany). The average potential of the mastoid electrodes (TP9 and TP10) served as the new reference. The main interference at 50 Hz was removed. The data were low-pass filtered at 30 Hz and high-pass filtered at 0.1 Hz (the slope was 24 dB/octave). Eye movement artifacts were corrected through independent component analysis (ICA) in Analyzer software, followed by visual inspection to ensure data quality. The EEG data were segmented for each participant and each condition separately. The length of the window was 1,250 ms, from 450 ms before and 800 ms after the target stimulus onset. A baseline correction was performed using data from −450 to −250 ms (*Kiefer & Martens, 2010*; *Ortells et al., 2016*). To ensure a stable and uncontaminated baseline reflective of a true task-free neural state, baseline correction was performed using a 200 ms window extracted from the 750 ms fixation period, rather than the subsequent 200 ms blank screen. EEG data exceeding ±100 µV were eliminated automatically as artifacts. Data from nine electrode sites, C1, C2, Cz, CP1, CP2, CPz, P1, P2, and Pz, were selected for the analysis of P3, with a time window of 300–600 ms. The P3 time window was defined based on both previous literature (*Wang, Jiao & Zhang, 2016*) and visual inspection of the grand-average ERP waveforms. The mean numbers of analyzable segments for athletes were 52.74 ± 6.89 in congruent conditions and 59.81 ± 5.92 in incongruent conditions. Similarly, non-athletes demonstrated comparable results, with mean analyzable segments of 55.21 ± 6.64 in congruent conditions and 62.84 ± 5.83 in incongruent conditions. Two-way repeated-measures ANOVAs were used to examine the mean peak amplitude and latency of P3.

Statistical analyses were conducted using SPSS 22.0, with a two-tailed significance level of $p < 0.05$. Despite the directional nature of our hypothesis, a two-tailed test was employed to ensure a conservative approach and to account for potential effects in both directions. The Greenhouse-Geisser method was used to correct for degrees of freedom and $p$ values for statistics that did not satisfy the test of sphericity. *Post hoc* tests were performed using the least significant difference method.

## RESULTS

### Identification rates

In the subjective threshold test, all participants reported that they could not consciously perceive the direction in which the prime stimuli were pointing. In the objective threshold test, one sample *t-tests* were performed separately on the $d'$ values of athletes and non-

athletes. The results showed that $d'$ in the athletes group was −0.06, which was not significantly different from chance ($t_{(19)}$ = −0.33, $p$ = 0.74). Similarly, $d'$ in the non-athletes group was −0.18, which was also not significantly different from chance ($t_{(20)}$ = −1.46, $p$ = 0.16). In addition, an independent sample $t$-test showed that there was no significant difference in $d'$ values between athletes and non-athletes ($t_{(39)}$ = 0.53, $p$ = 0.60).

### Response times and response errors

Independent t-tests were conducted to compare the IES of athletes and non-athletes across congruent and incongruent conditions. In the congruent condition, the mean IES for athletes was 602.23 ± 28.13, while for non-athletes it was 598.90 ± 28.82, with no significant difference observed between two groups ($t_{(39)}$ =−0.08, $p$ = 0.94). Similarly, in the incongruent condition, the mean IES for athletes was 427.19 ± 17.63, compared to 451.46 ± 36.16 for non-athletes, again showing no significant difference between two groups ($t_{(39)}$ = 0.59, $p$ = 0.56).

A two-way repeated-measures ANOVA for response times revealed a significant main effect of group ($F_{(1,39)}$ = 4.31, $p$ = 0.04, $\eta_p^2$ = 0.1). *Post-hoc* comparisons using Fisher's LSD test revealed that the response time in the athletes group was significantly less than that in the non-athletes group (athletes, 401.14 ± 9.69 ms; non-athletes, 429.26 ± 9.46 ms). Additionally, a significant main effect of response congruency was observed ($F_{(1,39)}$ = 68.75, $p$ = 0.00, $\eta_p^2$ = 0.64), indicating that participants responded faster on incongruent trials than on congruent trials (congruent, 449.92 ± 10.08 ms; incongruent, 380.47 ± 5.01 ms). The interaction between group and response congruency was not statistically significant ($F_{(1,39)}$ = 1.80, $p$ = 0.19, $\eta_p^2$ = 0.04) (Fig. 2).

A two-way repeated-measures ANOVA with error rates revealed a significant main effect of response congruency ($F_{(1,39)}$ = 32.9, $p$ = 0.00, $\eta_p^2$ = 0.46). *Post-hoc* comparisons show that participants committed fewer error on incongruent trials than on congruent trials (congruent, 22.77 ± 2.38%; incongruent, 9.95 ± 2.22%). The main effect of group was not significant ($F_{(1,39)}$ = 1.33, $p$ = 0.26, $\eta_p^2$ = 0.03). The interaction between group and response congruency was also not statistically significant ($F_{(1,39)}$ = 1.80, $p$ = 0.21, $\eta_p^2$ = 0.04) (Fig. 2).

### Correlation between years of training and the magnitude of NCE in table tennis athletes

The results of the Pearson's correlation analysis indicated a significant negative correlation between the years of training and the magnitude of NCE ($r_{(20)}$ = −0.48, $p$ = 0.03). This finding suggests that table tennis athletes with longer training experience exhibit a better response inhibitory control for unconscious visual-spatial information processing (Fig. 3).

### Electrophysiological results

#### P3 latency

A three-way repeated-measures ANOVA assessing P3 latency indicated a significant main effect of response congruency ($F_{(1,39)}$ = 170.42, $p$ = 0.00, $\eta_p^2$ = 0.81). *Post hoc* comparisons

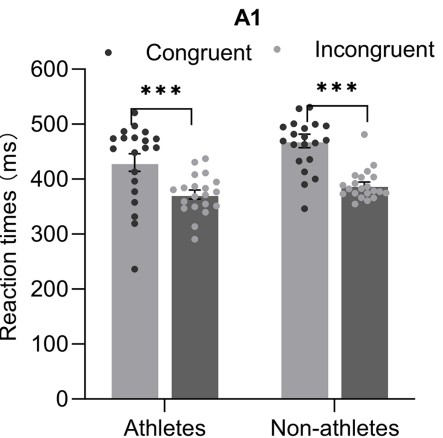
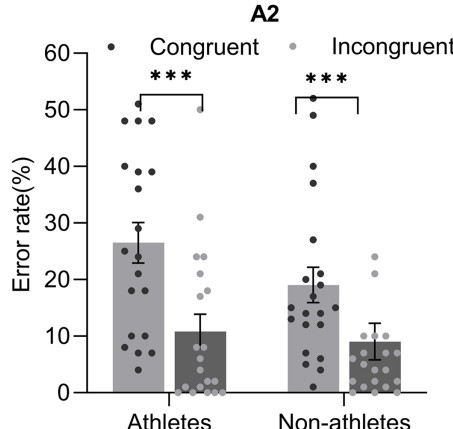

**Figure 2 Response times (A1) and error rates (A2) under congruent and incongruent conditions on the masked priming task for athletes and non-athletes.** Data represent mean ± standard error; ***$p$ < 0.001.                                 

indicated that the P3 latency on congruent trials was significantly longer than that on incongruent trials (congruent, 491.35 ± 8.34 ms; incongruent, 396.02 ± 8.35 ms). The main effect of electrode site was also significant ($F_{(8, 32)}$ = 5.58, $p$ = 0.00, $\eta_p^2$ = 0.13). *Post hoc* comparisons indicated that the latency was longest at the central site, and shortest at the parietal site.

The interaction between group and electrode site was significant ($F_{(8,32)}$ = 4.78, $p$ = 0.01, $\eta_p^2$ = 0.11). Simple effects analysis results showed that the latency at the central electrode site among athletes was greater than that among non-athletes ($p$ < 0.05); however, no significant difference was found in the latency of central-parietal and parietal electrode sites between these two groups ($p$ > 0.05) (Table 2).

The main effect of group was not statistically significant ($F_{(1,39)}$ = 3.22, $p$ = 0.08, $\eta_p^2$ = 0.08) nor was the interaction between group and response congruency ($F_{(1,39)}$ = 2.26, $p$ = 0.14, $\eta_p^2$ = 0.06). The interaction between response congruency and electrode site was not statistically significant ($F_{(8,32)}$ = 0.51, $p$ = 0.66, $\eta_p^2$ = 0.01). The interaction between group, response congruency, and electrode site was also not statistically significant ($F_{(8,32)}$ = 0.88, $p$ = 0.44, $\eta_p^2$ = 0.02) (Fig. 4).

### P3 peak amplitude

A three-way repeated-measures ANOVA with P3 amplitude revealed a significant main effect of group ($F_{(1,39)}$ = 8.72, $p$ = 0.01, $\eta_p^2$ = 0.18). *Post hoc* comparisons indicated that the amplitude among athletes was significantly higher than that among non-athletes (athletes, 10.40 ± 0.99 μV, non-athletes, 6.30 ± 0.97 μV). The main effect of electrode site was also significant ($F_{(8,32)}$ = 5.58, $p$ = 0.01, $\eta_p^2$ = 0.13). *Post hoc* comparisons indicated that the central site showed the highest amplitude, and the parietal site showed the smallest amplitude.

The interaction between group and electrode site was significant ($F_{(8,32)}$ = 3.59, $p$ = 0.04, $\eta_p^2$ = 0.08). Simple effects analysis results showed that the amplitudes of the central and

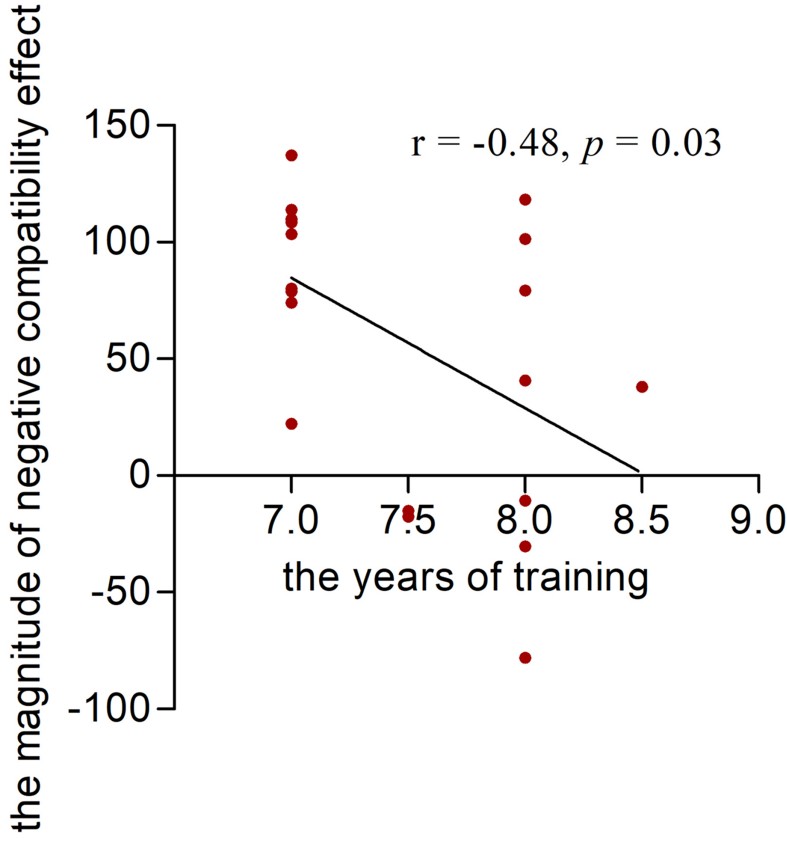

**Figure 3 The correlation between years of training and the magnitude of negative compatibility effect (NCE) in table tennis athletes.** Correlation between years of training and the magnitude of NCE in table tennis athletes. The abscissa shows the years of training, and the ordinate indicates the magnitude of NCE.

central-parietal sites among athletes were greater than those of non-athletes ($p < 0.05$); however, no significant difference was found between these two groups for the amplitude of parietal site ($p > 0.05$).

The interaction between response congruency and electrode site was significant ($F_{(8,32)} = 4.24$, $p = 0.01$, $\eta_p^2 = 0.10$). Simple effects analysis results showed that the amplitude of central site was greater than that of the central-parietal site in both congruent and incongruent conditions ($p < 0.01$), and the amplitude of central-parietal site was also greater than that of parietal site in both congruent and incongruent conditions ($p < 0.01$) (Table 3).

The main effect of response congruency was not significant ($F_{(1,39)} = 0.06$, $p = 0.80$, $\eta_p^2 = 0.00$) and the interaction between group and response congruency was not statistically significant ($F_{(1,39)} = 0.84$, $p = 0.37$, $\eta_p^2 = 0.02$). Similarly, the interaction between group, response congruency, and electrode site was not statistically significant ($F_{(8,32)} = 1.78$, $p = 0.16$, $\eta_p^2 = 0.04$) (Fig. 4).

**Table 2 P3 latency under congruent and incongruent conditions for athletes and non-athletes at each electrode site (mean ± standard error [ms]).**

| Electrode sites | Athletes | | Non-athletes | |
|---|---|---|---|---|
| | Congruent | Incongruent | Congruent | Incongruent |
| C1 | 509.60 ± 12.33 | 427.40 ± 13.40 | 485.62 ± 12.04 | 379.14 ± 13.08 |
| C2 | 511.50 ± 12.89 | 433.30 ± 13.23 | 487.14 ± 12.57 | 380.38 ± 12.91 |
| Cz | 508.00 ± 12.73 | 435.80 ± 13.45 | 484.19 ± 12.42 | 373.43 ± 13.13 |
| CP1 | 500.10 ± 12.09 | 418.90 ± 13.20 | 493.71 ± 11.80 | 380.57 ± 12.88 |
| CP2 | 511.50 ± 13.66 | 418.20 ± 13.41 | 484.48 ± 13.33 | 380.67 ± 13.08 |
| CPz | 503.40 ± 12.56 | 422.50 ± 14.31 | 485.71 ± 12.26 | 381.52 ± 13.96 |
| P1 | 491.90 ± 13.40 | 398.00 ± 11.84 | 485.81 ± 13.07 | 374.48 ± 11.55 |
| P2 | 486.70 ± 12.62 | 393.40 ± 12.05 | 483.62 ± 12.31 | 382.86 ± 11.75 |
| Pz | 487.60 ± 12.65 | 396.30 ± 11.67 | 483.43 ± 12.34 | 374.57 ± 11.39 |

## DISCUSSION

This study investigated the characteristics of unconscious information processing and its associated brain activity in general contexts. Cognitive processing was assessed between table tennis athletes and non-athletes during the completion of a masked priming task. Our evaluation of the participants' awareness of the priming stimulus from both subjective and objective aspects indicated that none of participants consciously perceived the priming stimulus. Numerous studies with the masked priming paradigm have found that a positive compatibility effect is observed at inter-stimulus interval (ISI) of 0–60 ms, whereas a negative compatibility effect occurs when the ISI is approximately 100–200 ms (*Sumner, 2008*; *Xia, Jiang & Zhang, 2013*). In the present study, we observed no positive compatibility effect-consistent with previous findings at an ISI of 33 ms, but instead found a negative compatibility effect, likely driven primarily by mask-induced inhibition mechanisms. The superimposed double arrows mask created directional conflict with the prime stimulus, triggering dual suppression: (a) the visual system classified the mask's bidirectional information as task-irrelevant noise, inducing global motor suppression that overrode the prime's activation; (b) structural similarity between mask and prime induced lateral inhibition in early visual pathways, selectively suppressing the prime's directional information through competitive interactions (*Schlaghecken et al., 2008*). The mask contained intrinsic directional information that directly conflicted with the prime's motor preparation, driving the motor system into a net inhibitory state. The mask's antagonistic motion signals systematically suppressed the very motor pathways pre-activated by the prime, resulting in the observed negative compatibility effect (NCE) (*Jaśkowski et al., 2008*; *Verleger et al., 2004*), manifesting as slower responses and higher error rates in congruent trials compared to incongruent trials (*Kahan & Chokshi, 2013*; *Sumner, 2008*). These findings indicate that the priming effect is mediated by the features of mask rather than solely by their presence.

Behavioral results showed that the response time of athletes was significantly shorter than that of the non-athletes, replicating the well-established speed advantage observed in

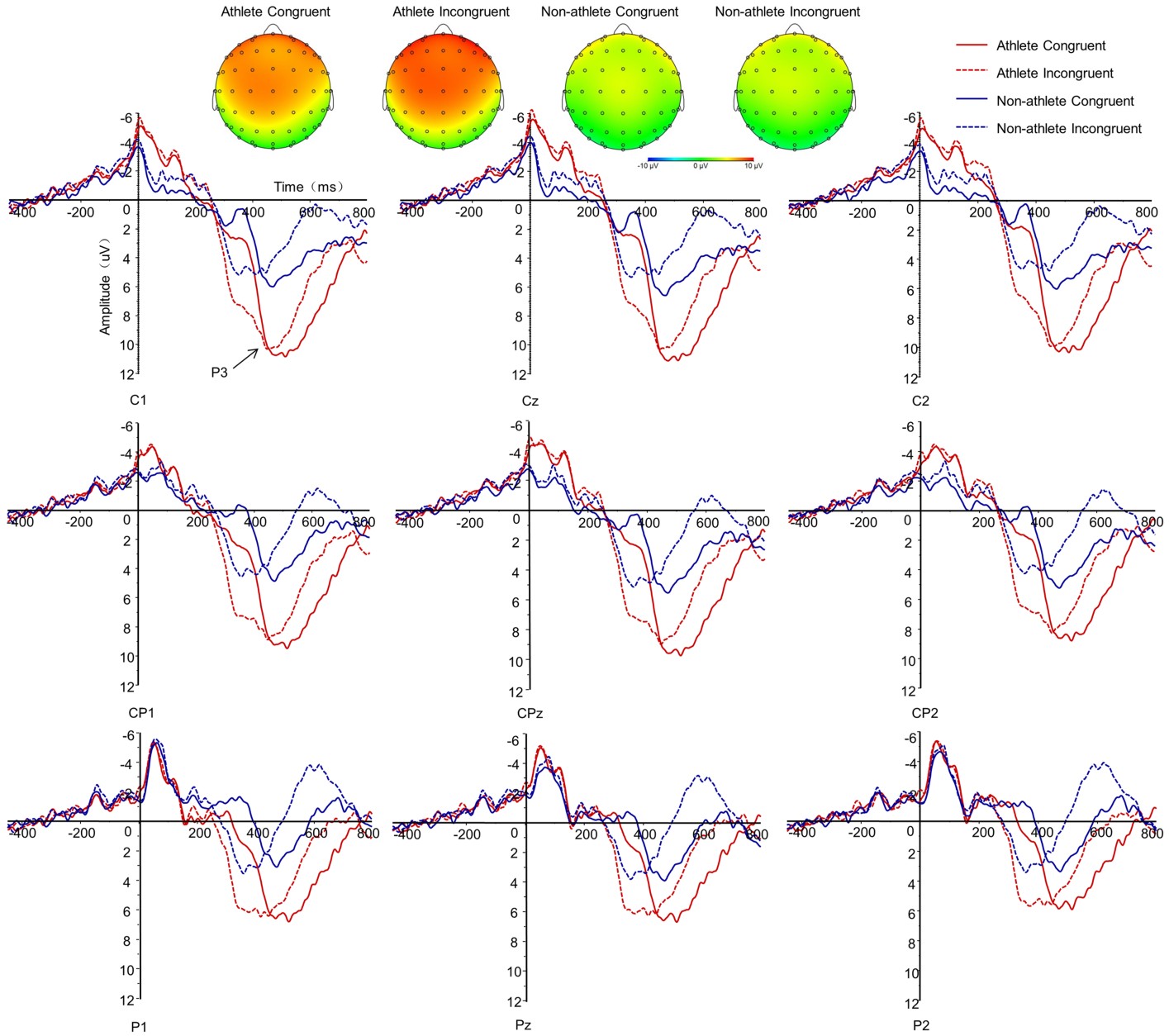

**Figure 4 The P3 ERPs elicited by the prime-target stimuli.** Topographic voltage maps of the peak amplitude of P3 for the congruent and incongruent conditions among athletes and non-athletes (upper panel). Grand average waveforms for the congruent and incongruent conditions among athletes and non-athletes at each electrode (lower panel).

previous studies (*Jiang, Xie & Li, 2021*; *Simonet et al., 2022*). Furthermore, we observed a significant negative correlation between years of training and the magnitude of NCE. Specifically, table tennis athletes with longer training exhibited smaller NCE, suggesting more efficient inhibitory control (*Zhu et al., 2022*). This implies that expertise facilitates rapid detection of prime-target conflict and more efficient suppression of competing

**Table 3 P3 peak amplitudes under congruent and incongruent conditions at various electrode sites for athletes and non-athletes (mean ± standard error [μV]).**

| Electrode site | Athletes | | Non-athletes | |
|---|---|---|---|---|
| | Congruent | Incongruent | Congruent | Incongruent |
| C1 | 12.33 ± 1.16 | 12.58 ± 1.19 | 7.87 ± 1.13 | 6.95 ± 1.16 |
| C2 | 12.01 ± 1.10 | 12.22 ± 1.13 | 7.80 ± 1.08 | 6.50 ± 1.10 |
| Cz | 12.82 ± 1.23 | 12.78 ± 1.24 | 8.37 ± 1.20 | 7.12 ± 1.21 |
| CP1 | 10.86 ± 1.08 | 10.96 ± 1.06 | 6.25 ± 1.06 | 6.13 ± 1.03 |
| CP2 | 10.29 ± 1.00 | 10.44 ± 1.07 | 6.74 ± 0.98 | 5.99 ± 1.05 |
| CPz | 11.18 ± 1.11 | 11.21 ± 1.10 | 7.03 ± 1.08 | 6.76 ± 1.08 |
| P1 | 7.98 ± 1.05 | 8.39 ± 0.95 | 4.61 ± 1.02 | 5.02 ± 0.92 |
| P2 | 7.00 ± 1.09 | 7.81 ± 1.02 | 4.77 ± 1.07 | 4.86 ± 1.00 |
| Pz | 7.89 ± 1.09 | 8.38 ± 1.04 | 5.40 ± 1.06 | 5.29 ± 1.01 |

motor responses (*Jiang, Xie & Li, 2024*; *Simonet, Beltrami & Barral, 2023*). However, our results showed that both athletes and non-athletes exhibited a robust NCE, which contrast with prior research on table tennis athletes. Previous studies have reported that table tennis athletes have a considerable advantage in process priming stimuli actively and efficiently at an unconscious level in both the sports-specific (*Meng et al., 2019a*; *Meng, Geng & Li, 2022*; *Shi et al., 2024*) and general contexts (*Geng et al., 2020*). The absence of unconscious processing advantage among table tennis athletes in present study may be attributable to: (a) contextual differences. Sports-specific and general stimulation scenarios are typically used in cognitive processing research for athletes. Compared with non-athletes, athletes exhibited higher perceptual sensitivity to sports-specific stimulation due to years of specialized experience or training (*Meng et al., 2019a*; *Meng, Geng & Li, 2022*), enabling quicker and more effective decisions under time pressure. In this study, double arrows-simple, widely recognized directional symbols- were used as the priming and target stimuli. Given their non-sport-specific nature, both athletes and non-athletes have high perceptual sensitivity. Even when the neural activation induced by the arrows below conscious awareness, participants could still perceive the directional information embedded in the primes. (b) perception-action coupling. According to event coding theory (*Hommel et al., 2001*), perception and action share a common representational system. Research demonstrates that this perception-action co-representation enables rapid responses at unconscious level (*Meng et al., 2019a*; *Meng, Geng & Li, 2022*). Through prolonged specialized training, table tennis athletes develop robust unconscious perception-action mappings for sport-specific stimuli, enabling automatic activating of corresponding motor responses (*Meng, Geng & Li, 2022*). However, both athletes and non-athletes showed high perceptual sensitivity to the generic arrow stimuli in the current study, we suggest that an unconscious link between the arrow stimuli and the corresponding responses had been established for both the athletes and non-athletes through everyday experience. The unconscious processing of the arrow stimuli

automatically triggered the corresponding response regardless of motor expertise. This may explain the comparable negative compatibility effects observed between groups. (c) task difficulty. As evidenced by *Liu et al. (2022)*, cognitive task difficulty may be an important factor influencing athletes' performance. For instance, *Geng et al. (2020)* demonstrated superior unconscious information processing within general context. Differences in task difficulty may account for the discrepancies between our findings and those of *Geng et al. (2020)*. In contrast to *Geng et al.*'s *(2020)* design, which featured four combinations of non-directional priming and target stimuli, our protocol was limited to two directional priming-target combinations, potentially reducing task difficulty. Therefore, we hypothesize that the lack of evident superiority in athletes' unconscious information processing may reflect a ceiling effect in low-difficulty task, with their advantages becoming more pronounced under higher cognitive demands.

ERP results showed that a robust NCE in both groups, with longer P3 latencies in the congruent condition than in the incongruent condition, indicating delayed inhibitory control processing. Additionally, the ERP analysis revealed no interaction between group and response congruency for either P3 latency or peak amplitude, suggesting that motor expertise does not significantly modulate either the timing or intensity of neural responses to response conflict. The presence of NCE across both behavioral and EEG measures confirms that unconscious processing of priming stimuli occurs similarly in both athlete and non-athlete in the general contexts, replicating previous findings (*Shi et al., 2024*; *van Gaal et al., 2011*). Critically, the ERP analysis demonstrated that P3 latency of the athletes was significantly longer than that of the non-athletes in the central region of the brain and that the peak P3 amplitude among athletes was significantly higher than that of among non-athletes in the central and central-parietal brain regions. These findings suggested that the delay onset of inhibitory control and augmented cortical engagement. We propose that this phenomenon may reflect neurocognitive adaptations resulting from prolonged sport-specific table tennis training. Table tennis, characterized by its high speed and variability, requires athletes to process multiple types of information and execute decisions within milliseconds (*Bootsma & van Wieringen, 1990*; *Guo et al., 2020*; *Zhang et al., 2020*). Extended training in such open-skill sports has been shown to induce structural and functional modifications in the fronto-basal response control network (*Chavan et al., 2017*), potentially enabling lower perceptual thresholds for fast information (*Meng, Geng & Li, 2022*). While these adaptations optimize athletic performance, they generalize to non-sport contexts, leading to deeper information processing automatically. This elevated cognitive engagement underlies the prolonged P3 latency observed in table tennis athletes. Notably, athletes invested more attention resources to enhance the accuracy of their responses when the prime-target required divergent responses even at unconscious level. Extensive table tennis training induces neuroplasticity, refining cognitive resource allocation during perceptual processing. Despite inhibitory control processing being somewhat later in athletes, they were still able to inhibit the tendency for erroneous movement and complete the adjustment of the movement response so as to respond quickly and efficiently in the congruent condition (*Chen et al., 2019*; *You et al., 2018*).

The findings of this study carry significant implications for both sports training and cognitive enhancement. The unconscious inhibitory control exhibited by high-level table tennis athletes indicates that specialized training may bolster cognitive control and decision-making capabilities under time pressure, primarily within the context of the specific sport. These insights could guide the creation of targeted training regimens aimed at elevating athletic performance. Furthermore, the benefits of specialized training methodologies might extend the realm of sports, potentially aiding individuals in high-stakes professions, such as emergency responders or military personnel, who must make rapid and accurate decisions in pressurized scenarios.

Although our study provides valuable insights concerning the unconscious information processing of table tennis athletes in general contexts, it also has several limitations. Firstly, one limitation of this study is the relatively small sample size, particularly the limited number of higher level table tennis athletes, which could reduce statistical power and increase the risk of Type II errors. Furthermore, the sample was not sufficiently large to examine potential sex differences. Future research should aim to include a larger, more balanced cohort of national-level table tennis athletes to both enhance the robustness of the findings and enable meaningful investigation of sex-specific effects in table tennis performance. Secondly, the fitness levels of participants in the two groups were not assessed or matched. Given that fitness levels can significantly influence cognitive performance, this could have introduced a potential confounding factor that skew our results. Future studies should consider measuring and controlling the fitness levels to ensure more robust comparisons between groups. Thirdly, we did not monitor error rates or post-error slowing, which are important indicators of cognitive control processes. Future research should consider incorporating error-related brain activity and post-error behavioral adjustments to further explore the dynamics of cognitive control. Finally, a proportion of elite table tennis athletes are left-handed, our study exclusively recruited right-handed athletes, potentially limiting the generalizability of our findings to the broader athlete population. Future studies should incorporate left-handed table tennis athletes to better explain the sport-specific neuroadaptations.

## CONCLUSIONS

The results of this study investigating the characteristics of unconscious information processing and its associated brain activity in the general contexts among table tennis athletes performing a masked priming task suggest that unconscious information processing is not prominent in the general context, but may be limited to the sports-specific context or accessed during more complex cognitive tasks. Although table tennis athletes showed no advantage for unconscious information processing in the general context, they showed better performance than non-athletes in the masked priming task by flexibly allocating attention resources and efficiently modulating action processing, benefits from their specialized training experience.

### Funding

This research was funded by the Humanities and Social Sciences Pre-research Project of Huzhou University (2020SKYY11). The funders had no role in study design, data collection and analysis, decision to publish, or preparation of the manuscript.

### Grant Disclosures

The following grant information was disclosed by the authors:
Humanities and Social Sciences Pre-research Project of Huzhou University: 2020SKYY11.

### Competing Interests

The authors declare that they have no competing interests.

### Author Contributions

- Fanying Meng conceived and designed the experiments, performed the experiments, analyzed the data, prepared figures and/or tables, authored or reviewed drafts of the article, and approved the final draft.
- Lijiao Chen conceived and designed the experiments, performed the experiments, authored or reviewed drafts of the article, and approved the final draft.
- Chun Xie performed the experiments, prepared figures and/or tables, and approved the final draft.
- Jiadong Zheng analyzed the data, prepared figures and/or tables, and approved the final draft.
- Ning Chen analyzed the data, authored or reviewed drafts of the article, and approved the final draft.
- Fanghui Qiu conceived and designed the experiments, prepared figures and/or tables, and approved the final draft.
- Jiaxian Geng conceived and designed the experiments, performed the experiments, analyzed the data, prepared figures and/or tables, authored or reviewed drafts of the article, and approved the final draft.

### Human Ethics

The following information was supplied relating to ethical approvals (*i.e.*, approving body and any reference numbers):

This study protocol received approval from the Ethics Committee of Huzhou University (No. 20200528-SGY13).

### Data Availability

The data are available in the Supplemental File.

## Supplemental Information

Supplemental information for this article can be found online at http://dx.doi.org/10.7717/peerj.19508#supplemental-information.

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
