# Peer review of "Unconscious information processing of table tennis athletes in a masked priming paradigm: an event-related potentials (ERP) study"

_PeerJ, doi:10.7717/peerj.19508_

## Round 0.1 · original submission · Major Revisions

As you will see, the reviewers are largely supportive of progressing this paper, but they agree that while your manuscript addresses an important topic, several methodological shortcomings require substantial attention to strengthen the validity and reproducibility of its findings. This must be corrected and addressed if we are to progress the paper. In particular, the reviewers raise significant concerns about the methodological validity of the study. Issues they raise include a lack of clarity in the study design, particularly discrepancies in task timing and insufficient details on the inter-trial interval, recruitment criteria, and participant characteristics. The study appears underpowered for the analyses performed, and critical confounders such as fitness levels, medication use, and psychopathological history were not controlled, which could undermine the reliability of the findings. Details about EEG data acquisition and processing, including artefact handling, epoch balancing, and justification for baseline and component time windows, are insufficient, limiting replicability. Behavioural analyses also lack methods to address response time and accuracy trade-offs, such as inverse efficiency scores. The reviewers question the interpretation of neural markers like P3 amplitude, suggesting alternative explanations such as neural efficiency, and recommend examining training duration and sex differences to better contextualise the results.

·

Basic reporting

Interesting idea of ​​this study, my recommendations are the following:
Abstract – I recommend mentioning the average age and standard deviation of the subjects in the two groups. In the Methods section I recommend mentioning the name of the equipment used in the evaluation. I recommend correcting the final number of subjects participating in the study, not recruited, according to the Participants section.
Introduction section – I recommend expanding it by mentioning relevant aspects regarding the topic of the study specific to table tennis.
Section 2 Materials and Methods – I recommend introducing a new subsection called Study design where to mention the typology of the study and other specific aspects.
At the end of the Discussion section I recommend mentioning future research directions. I recommend mentioning the practical implications of the present study.

Experimental design

The study is well organized, with detailed presentation of both the testing instruments and the procedure. The results are reliable, consistent, and well interpreted without duplication of information.
Section 2 Materials and Methods – I recommend introducing a new subsection called Study design where to mention the typology of the study and other specific aspects.

Validity of the findings

The results are reliable, consistent, and well interpreted without duplication of information.

Additional comments

-

Reviewer 2 ·

Basic reporting

The manuscript presents the results of a study in which a masked priming paradigm with simultaneous EEG recording was used to examine unconscious information processing and its neural correlates in high-performance athletes. Overall, the manuscript is clearly written and conforms to scientific standards, although it has formal, conceptual, and technical limitations, which are described below.
The manuscript is well-structured, with the expected sections. The tables and figures are relevant and informative, although Fig. 2 does not follow the recommendation to overlay bar graphs with scatter plots showing individual data points or use another method to show the distribution of the data. Although the files that the authors submitted with the tables and figures do not contain the titles, these have been made available on the submission platform and seem to adequately describe the respective content. All appropriate raw data have been made available. Conversely, human participants approval documentation and consent form were submitted in Chinese only, making it impossible to verify their adequacy.
To conclude this initial and more general section of the basic reporting, authors are advised to thoroughly check in-text citations and references. There are numerous formatting inaccuracies (e.g., no comma needed before et al.), typos (e.g., "Schlaghecken, te al., 2008", missing or unnecessary spaces), year discrepancies (e.g., see "Gilovich, Griffin & Kahneman, 2006" or "De Pisapia, 2004"), author mismatches (e.g., see "Kisel, et al., 2009"), and reference entries not cited in the text (e.g., "Jiang, R., Xie, F., & Li, A. (2024)"). There are also some words that are unnecessarily hyphenated (e.g., “com-pared” or “in-formation”).
Moving on to the introduction, this section of the ms presents relevant previous literature to contextualize the study and how it fits with existing research in the field. However, some sentences seem a bit loose in the narrative, with redundancies that could be mitigated (e.g., the idea that athletes process unconscious information more effectively is overly repeated), and central aspects of the study are not sufficiently supported in the background. For example, the interest in studying table tennis players rather than athletes from other sports is tacitly understood. However, this option is not adequately supported. Similarly, the framework presented is not sufficient to support the study of the negative rather than the positive compatibility effect. In fact, the short time-windows (SOA) from prime to target stimulus suggest that the authors were interested in the typical positive compatibility effect. If this was not the case, then the paradigm could have been better designed (authors seem to indirectly acknowledge this issue in lines 381-385). Trusting that the hypothesis was defined a priori, even though the paradigm is not optimized for the negative compatibility effect, it requires further specification, namely, it would be important to specify how the athletes' better performance was expected to translate into behavioral results and their neural correlates.
There is also a basic conceptual or terminological misunderstanding in the manuscript: the fact that arrows are used instead of sport-related stimuli (e.g., rackets and balls) does not mean that general instead of specific cognitive domains are involved. The term general cognitive domain refers to versatile mental processes that are broadly applicable across various cognitive activities. Regardless of whether rackets or arrows are used as stimuli, the cognitive processes that are involved in the task remain the same: selective attention, cognitive control, response inhibition, conflict monitoring, and processing speed. These are typical processes of the general cognitive domain. In other words, even though a racket is a sport-specific stimulus, using it does not necessarily induce a specific cognitive process. What the authors managed to investigate was simply whether the results of previous studies with one type of stimulus could be generalized or at least transferred to another type of stimulus, while the underlying cognitive processes remain the same.
Regarding methods and results sections, although some of the statistical tests performed were not aimed at testing the initial hypothesis (e.g., electrode-site effects), the results are clearly reported, regardless of some formatting inconsistencies (e.g., use of italics in statistical symbols; use space before/after mathematical operators) and insufficient statistical detail in some cases, as noted below.
Finally, the discussion is presented in an appropriate tone and in line with the results, which mostly concern null effects. However, some sentences are puzzling, for example: “the P3 latencies were shorter in the incongruent condition than in the congruent condition, indicating that inhibitory control processing appeared later in the incongruent compared with the congruent condition.”

Experimental design

With regard to the experimental, there are minor inconsistencies and the information provided is not sufficient to properly ensure replication. Specifically:
• While the text reports that the blank screen was displayed for 1000 ms, the diagram shows 200 ms;
• If participants were instructed to respond to the target in 1100 ms and the blank screen that followed was displayed for 200 ms, it is not clear what the participants were observing in the response period;
• It is also unclear what the intertrial interval refers to (it is not in the diagram), i.e., whether it includes the blank screen and extends to the fixation cross, or whether it follows the blank screen (in which case, what were participants observing during the intertrial interval?);
• For rigor, the electrode configuration used is an extension of the international 10-20 system (64-electrode caps typically use 10-10 IS);
• For the sake of replication, it is important to provide additional technical details (model and brand of EEG caps, model and brand of amplifier, filtering during acquisition, etc.) - authors are encouraged to consider the paper by Picton et al. (2000) on Guidelines for Using Human Event-Related Potentials to Study Cognition: Recording Standards and Publication Criteria (DOI:10.1017/S0048577200000305);
• Similarly, it is unclear whether the epochs relating to incorrect or at least missed trials were removed from the ERP analyses and whether the number of epochs was adequately balanced between experimental conditions (namely after artifact correction/removal procedures);
• ICA and a 100 μV threshold were used to remove eye movement artifacts and presumably blinks, but nothing is said about other possible artifacts and removal/correction procedures (e.g., was visual inspection used?);
• The BL time-window is not justified, especially taking into consideration that such a time-window captures the change between the fixation cross and the blank screen (without proper justification for the adopted BL, using the 200 ms of the latter could be a better option);
• The definition of the P3 time-window lacks further justification;
• Given that the hypothesis is clearly directional, the reason for using a two-sided statistical threshold (alpha; not p) is questionable;
• The type of post-hoc tests was not specified.

Validity of the findings

Not enough information is provided to accurately replicate the sample size estimation, but from the parameters provided it seems that the study is underpowered: a mixed RM ANOVA with two groups (athletes, non-athletes) and two measurements (congruent, incongruent) would require a larger sample.
Also, given the nature of the study, the sample could be better characterized in terms of variables with a potential confounding effect on behavioral and biological measures, such as medication and drug use, as well as a history of psychopathology or neuropathology. In addition, although the groups are likely to be matched for educational level, it would be useful to present this information, as education can improve processing speed and accuracy, and higher levels of education are often associated with better cognitive control and inhibition.
The validity and reliability of the findings are affected by the above-mentioned limitations and the lack of information needed to ensure reproducibility. Taking all of the above into account, in my view, the manuscript would need substantial improvement, and there are methodological issues that may be difficult to address in order to present robust findings, despite the scientific relevance of the topic and the merit of its study.

Additional comments

Having been a semi-professional table tennis player myself, I am intrigued by the characteristics of the TT players: professional players typically start practicing before the age of 10, about 25% of them are left-handed, and spend more than 20 hours a week training, characteristics that are not represented in the sample of athletes, despite being identified as players with professional training.

·

Basic reporting

In line 20, the method in the abstract, ““Twenty-two table tennis athletes (athletes) and 22 aged-matched” is the word “athletes (athletes)” in parentheses is redundant? In line 22, “task (A prime stimulus (arrows pointing left or right) was” is incorrectly bracketed. Line 24, “in the opposite direction for incongruent.) while .....” Same thing here, and incorrect use of periods. Line 266, “congruency (F(1,39) = 2.26, p = 0.14, = 0.06).” The parentheses after F are inconsistently sized. Line 333, “sensitivity to the double arrows used in this task; thus, even .....” There are many similar details, although they are all minor issues, they can cause difficulties for readers to read. At the same time, this also reflects the author’s seriousness towards the manuscript, so please check the manuscript carefully. In addition, there are some grammatical errors in some sentences in the manuscript, such as in line 112-113, “Based on the results of previous studies that found that specialized training.....”, in line 205-206 “with group (athlete vs. control)” it should be “with groups (athlete vs. control)”. In line 336 “A second plausible.... perception action coupling.” There are many other similar sentences in the manuscript, and the authors are requested to check the whole text carefully. The English language should be improved to ensure that an international audience can clearly understand your text.

Experimental design

The authors have presented the section on experimental design in a relatively detailed manner, both in terms of the use of tools and the specific process of the experiment. However, there are still some minor questions that need to be answered by the author. Firstly, what criteria does the author have for recruiting participants in the experiment? How were these table tennis players and non-athletes recruited? The author should display it in the manuscript. Second, in lines 156-157, “The target stimulus was then displayed for 100 ms, followed by another blank screen displayed for 1000 ms.” It does not appear to be seen in Figure 1 that after displaying the target stimulus for 100 ms, another blank screen is then displayed for 1000 ms. After 100 ms in Figure 1, a blank screen of 200 ms is displayed. Is this the reason for the image display? What else is the reason? Furthermore, in section 2.5 of the data analysis, the author utilized numerous subheadings. It is recommended that the author consolidate the related content, as there may not be a necessity for these subheadings.

Validity of the findings

Regarding the findings, the authors have presented and effectively analyzed the experimental result data. However, there are a few minor details that require the authors’ attention. In lines 269-270, “significant (F(8,32) = 0.88, p = 0.44, =0.02)(Table 2; Figure 3).” It is advisable not to position Table 2 and Figure 3 in the same spot simultaneously. Instead, they should be placed at the end where they best align with the text content. This approach helps ensure that readers can understand the information more easily. The same guideline applies to “(Table 3; Figure 3).” Additionally, the discussion section is overly lengthy and lacks close integration with the discussion of the original research question. It is recommended that the author reorganize this section and remove unnecessary content. Why does the author say “ We hypothesize that this finding was related to the characteristics of table tennis.” in discussion. Meanwhile, the discussion section needs to be reorganized and written with a better logical line.

Additional comments

Many punctuation marks in the manuscript are not used correctly, some subheadings are a bit unclear, and some are not. For example, in line 190, “2.5.1. Identification rates” No need to point after 1. The same applies to “2.5.2.”, “2.5.3.”, “3.1.”, “3.2.”, “3.3”, “3.3.1.”, “3.3.2”. Regarding the (p values<0.05) in the manuscript, if the author has a specific p value, it can be written directly or the vaults can be removed and changed to p<0.05. For abbreviations, in line 98, “Event-related potentials (ERPs) have” is ERPs, but then it is REP. Please note that there is a difference in the form of the abbreviations to maintain consistency in the abbreviations, even though they all mean the same thing. In terms of references, some journal names in the reference section use full names, while others use abbreviations, resulting in inconsistent forms. Additionally, according to the requirements of Peer J journal’s official website for reference format, the journal name needs to be the full name. The author is requested to carefully check the format of the references and make modifications. Furthermore, in the manuscript with four or more authors, abbreviate with ‘first author’ et al. (e.g. Smith et al., 2005). “(Wang, et al., 2024; Ziri, et al., 2024)” does not require a comma after first author name.

·

Basic reporting

no comment

Experimental design

no comment

Validity of the findings

no comment

Additional comments

This study investigates the unconscious information processing of table tennis athletes using a masked priming paradigm combined with EEG recording. While the study presents intriguing results, there are several methodological and analytical concerns that need to be addressed to strengthen the conclusions. Additionally, the manuscript could benefit from a more thorough integration of existing literature and clearer articulation of the study's novelty and implications.

1. The introduction provides a reasonable overview of unconscious information processing in athletes. However, it lacks a comprehensive discussion of the broader literature on perceptual and cognitive abilities in athletes. For example, the introduction could benefit from a broader discussion of prior studies on perceptual and cognitive abilities in athletes. Specific references such as Yao et al. (2020) and Meng et al. (2019) on sports expertise and cognitive processing should be incorporated for a more comprehensive literature review. Incorporating relevant references and highlighting the gap in knowledge addressed by the current study would strengthen the rationale. Here are some references that are in line with these topics:
1. Meng, F.-W., Yao, Z.-F., Chang, E. C., Chen, Y.-L. (2019). Team sport expertise shows superior stimulus-driven visual attention and motor inhibition. PLoS One.
2. Yao, Z.-F., Sligte, I. G., Moreau, D., et al. (2020). The brains of elite soccer players are subject to experience-dependent alterations in white matter connectivity. Cortex.
3. Chen, Y. L., Hsu, J. H., Tai, D. H., & Yao, Z. F. (2022). Training-associated superior visuomotor integration performance in elite badminton players. IJERPH.
4. Yao, Z.-F., Fu, H.-L., Liang, C.-W., et al. (2024). Electrophysiological differences in inhibitory control processing between collegiate-level soccer players and non-athletes. Brain and Cognition.
5. Yao, Z. F., Sligte, I. G., & Ridderinkhof, R. (2024). Olympic team rowers and team swimmers show altered functional brain activation during working memory and action inhibition. Neuropsychologia, 108974. Advance online publication. https://doi.org/10.1016/j.neuropsychologia.2024.108974
2. The novelty of the method and how it addresses gaps in the literature is not well articulated. This should be clarified to highlight the unique contributions of the study.
3. It is commendable that the authors matched the groups for demographic variables; however, fitness levels, a critical factor influencing physiological signals and performance, were not controlled. Fitness levels should be assessed or discussed as a potential confound.
4. Were sex differences considered? If not, this could limit the generalizability of the findings. A stratified analysis based on sex might be warranted.
5. Balancing speed and accuracy is essential in cognitive tasks. Please elaborate on the methods used to address the RT-accuracy trade-off in your behavioral analyses. Techniques such as the computation of inverse efficiency scores or the application of the drift-diffusion model can provide insights into participants' performance strategies.
6. The duration between the prime and target (SOA) in masked priming paradigms can influence cognitive processing. Please clarify whether you manipulated the SOA in your study and provide a rationale for the chosen durations. Varying SOAs can help disentangle automatic from controlled processing mechanisms.
7. The P3 component is associated with theta oscillations and can be subdivided into P3a and P3b, each reflecting different cognitive processes. Did your analysis consider these subcomponents and their relation to preceding components like N2? Additionally, did you perform time-frequency analyses to examine theta power modulations? Such analyses can provide deeper insights into the neural dynamics underlying cognitive control in athletes.
8. It's important to assess whether non-athletic control participants have any systematic training experiences that could influence cognitive performance. Please specify the criteria used to select control participants and whether their training backgrounds were considered to ensure a valid comparison with athlete groups.
9. It would be valuable to investigate the correlation between years of training and ERP components (e.g., P3 latency, amplitude) to determine whether longer training durations are associated with stronger neural markers of cognitive control or enhanced unconscious processing.
10. While your findings indicate greater P3 amplitude in athletes, which you interpret as increased attentional resource allocation, consider discussing whether this reflects "neural efficiency" (i.e., performing the task with fewer resources). This aligns with previous sports neuroscience research, where enhanced efficiency is often observed in highly skilled athletes.
11. Were other task parameters (e.g., error monitoring or post-error slowing) considered? Including such variables may enrich the discussion of cognitive control.
12. In your discussion of the NCE, consider elaborating on whether the NCE observed in athletes reflects strategic suppression of automatic responses or a byproduct of priming effects. This could be supported by citations of studies that distinguish between perceptual inhibition and motor response inhibition.

---

## Round 0.2 · accepted · Accept

Thank you for addressing the reviewers' original comments. This work is now suitable for publication.

·

Basic reporting

The authors revised the article in accordance with all recommendations. I have no further recommendations to suggest to the authors.

Experimental design

The authors have revised the article in accordance with all recommendations. I have no further recommendations to suggest to the authors.

Validity of the findings

The authors have revised the article in accordance with all recommendations. I have no further recommendations to suggest to the authors.

Additional comments

-

·

Basic reporting

no comment

Experimental design

no comment

Validity of the findings

no comment